# Dispersal dynamics of SARS-CoV-2 lineages during the first epidemic wave in New York City

Simon Dellicour[1,2]*, Samuel L. Hong[2], Bram Vrancken[2], Antoine Chaillon[3], Mandev S. Gill[2], Matthew T. Maurano[4,5], Sitharam Ramaswami[6], Paul Zappile[6], Christian Marier[6], Gordon W. Harkins[7], Guy Baele[2], Ralf Duerr[4,8]*, Adriana Heguy[4,6]*

1 Spatial Epidemiology Lab (SpELL), Université Libre de Bruxelles, Bruxelles, Belgium, 2 Department of Microbiology, Immunology and Transplantation, Rega Institute, Laboratory for Clinical and Epidemiological Virology, KU Leuven, University of Leuven, Leuven, Belgium, 3 Division of Infectious Diseases and Global Public Health, University of California San Diego, California, United States of America, 4 Department of Pathology, NYU Grossman School of Medicine, New York, New York, United States of America, 5 Institute for Systems Genetics, NYU Grossman School of Medicine, New York, New York, United States of America, 6 Genome Technology Center, Office for Science and Research, NYU Langone Health, New York, New York, United States of America, 7 South African Medical Research Council Capacity Development Unit, South African National Bioinformatics Institute, University of the Western Cape, Bellville, South Africa, 8 Department of Microbiology, NYU Grossman School of Medicine, New York, New York, United States of America

* simon.dellicour@ulb.ac.be (SD); ralf.duerr@nyulangone.org (RD); adriana.heguy@nyulangone.org (AH)

**Data Availability Statement:** R scripts and related files needed to run all the analyses are all available at https://github.com/sdellicour/sars-cov-2_new_york.

## Abstract

During the first phase of the COVID-19 epidemic, New York City rapidly became the epicenter of the pandemic in the United States. While molecular phylogenetic analyses have previously highlighted multiple introductions and a period of cryptic community transmission within New York City, little is known about the circulation of SARS-CoV-2 within and among its boroughs. We here perform phylogeographic investigations to gain insights into the circulation of viral lineages during the first months of the New York City outbreak. Our analyses describe the dispersal dynamics of viral lineages at the state and city levels, illustrating that peripheral samples likely correspond to distinct dispersal events originating from the main metropolitan city areas. In line with the high prevalence recorded in this area, our results highlight the relatively important role of the borough of Queens as a transmission hub associated with higher local circulation and dispersal of viral lineages toward the surrounding boroughs.

## Author summary

In the context of epidemics, analyses of viral genomes can be used to link infectious cases in space and time. When the sampling is dense enough, phylogeographic investigations can be performed to obtain estimates of the dispersal history and dynamics of viral lineages. In our study, we take advantage of a comprehensive data set of SARS-CoV-2 genomes sampled from New York State to analyze the circulation of the virus during

**Funding:** SD is supported by the Fonds National de la Recherche Scientifique (FNRS, Belgium). SLH acknowledges support from the Research Foundation - Flanders (Fonds voor Wetenschappelijk Onderzoek - Vlaanderen, G0D5117N). BV is supported by a post-doctoral research fellowship (grant nr. 12U7121N) of the Research Foundation - Flanders (Fonds voor Wetenschappelijk Onderzoek). AC was supported by grants from the NIH (San Diego Center for AIDS Research, CFAR, AI306214 and AI100665), and the James B. Pendleton Charitable Trust. MTM is supported by a NIH Grant (R35GM119703). GWH is jointly funded by the South African Medical Research Council and the National Institutes of Health, USA grant 1U01AI152151-01. GB acknowledges support from the Research Foundation - Flanders (Fonds voor Wetenschappelijk Onderzoek - Vlaanderen, G0E1420N, G098321N) and from the Interne Fondsen KU Leuven/Internal Funds KU Leuven under grant agreement C14/18/094. RD was partially supported by the NIH grant 1R01AI122953. AH, SR, CM, and PZ are supported by the Genome Technology Center, which is in part by the Cancer Center Support Grant P30CA016087 at the Laura and Isaac Perlmutter Cancer Center. The funders had no role in study design, data collection and analysis, decision to publish, or preparation of the manuscript.

**Competing interests:** The authors have declared that no competing interests exist.

spring 2020. In particular, we focus on New York City, then the epicenter of the United States epidemic, to unravel the dispersal of viral lineages among its five boroughs, which tends to confirm the relative importance of the Queens area in the overall transmission chain. From a methodological point of view, our study illustrates how fine-scale phylogeographic analyses can be exploited to gain insight into the epidemiological dynamic of local viral epidemics, which constitute timely but also complementary tool to standard epidemiological approaches.

## Introduction

During spring 2020, New York City (NYC) was the epicenter of the coronavirus disease 2019 (COVID-19) epidemic in the United States [1]. Despite some travel restrictions, the first COVID-19 positive case was identified in NYC on February 29, 2020 (a healthcare worker who would have contracted the virus while traveling in Iran), and was soon followed by the detection of community transmission of severe acute respiratory syndrome coronavirus 2 (SARS-CoV-2), its causative agent [2]. During the first three months of the NYC epidemic, more than 200,000 people tested positive for the virus and more than 18,500 COVID-19 deaths were reported [3].

Phylogenetic investigations indicated that early virus introduction events into NYC were most likely related to viral lineage importations from Europe and other US states [4]. However, while phylogenetic clusters reflecting local transmission have been identified in the metropolitan area [2,4], little is known about the actual circulation of viral lineages within NYC, a city that was severely but also heterogeneously impacted by the first epidemic wave, with the highest prevalence estimated for the borough of Queens [1]. Specifically, there has been a lack of fine-scale phylogeographic investigations that could help understand to what extent transmission clusters were constrained to local NYC areas or alternatively spread across the city. By placing phylogenetic trees in space and time [5], viral phylogeographic approaches constitute relevant tools to elucidate the dispersal dynamics of the first epidemic wave that hit NYC.

In the present study, we used viral genomic data generated by our study team and deposited on GISAID (NY-NYUMC2-929) to gain insights into the dispersal dynamics of SARS-CoV-2 within NYC, one of the main global epicenters during the first months of the pandemic. For this purpose, we applied continuous and discrete Bayesian phylogeographic approaches while accounting for sampling heterogeneity to avoid artifacts related to heterogeneous sampling efforts. Specifically, we aim to investigate to what extent viral lineages circulated within and among the five NYC boroughs (Manhattan, the Bronx, Brooklyn, Queens, and Staten Island).

## Results and discussion

In the first part of the study, we applied a rapid analytical pipeline in which phylogeographic reconstructions were performed along a fixed time-scaled phylogenetic tree, which reduces computational limitations associated with very large data sets [6]. We inferred such a time-scaled phylogeny based on 828 viral genomic sequences collected within New York State throughout the first epidemic wave (March-May, 2020), as well as all 1,899 background sequences used in the Nextstrain [7] build focused on North America (which also includes genomic sequences from the other continents), for a total of 2,727 SARS-CoV-2 genomic sequences. As described in the Methods section, the time-scaled phylogenetic tree was inferred following a procedure similar to the one used by the Nextstrain platform [7] (nextstrain.org),

and the resulting tree employed as a fixed topology in initial Bayesian discrete [8] and continuous [9] phylogeographic analyses.

The discrete phylogeographic analysis first aimed at detecting and quantifying lineage introduction events into New York State (S1 Fig), and identified 116 independent introductions (95% HPD interval = [107–127]) along GISAID clades O, S, V, G, GR, and GH, and a strong dominance of spike mutation D614G associated with G clades. Considering the number of sequences sampled from New York State ($n$ = 828), this estimated number of lineage introduction events quantifies the relative contribution of external introductions in establishing local transmission chains in the study area. As part of the 116 independent introduction events, our analysis identified a major clade connecting 588 GISAID clade GH sequences sampled in New York State, and which was used as a starting point for the phylogeographic analyses performed in the second part of the study (see below). Of the remaining introductions, the majority were represented by singletons (69%), or formed relatively small SARS-CoV-2 clades (S1 Fig), thus illustrating the great heterogeneity that exists in the ability of the virus to successfully establish ongoing local transmission chains. Furthermore, estimations of the time to the most recent common ancestor (TMRCA) of each clade indicate that those introduction events were not concentrated at the beginning of the epidemic (Fig 1), with a large proportion (83%) of TMRCAs falling in the second half of March, 2020. The subsequent continuous phylogeographic analyses aimed at inferring the lineage dispersal history of New York State clades delineated in the previous steps. Continuous phylogeographic reconstructions highlight that most peripheral samples are directly connected to the NYC area rather than clustered together (Fig 1). While this phylogeographic pattern would indicate that samples collected outside NYC likely correspond to distinct dispersal events originating from the city area, we however acknowledge that areas surrounding NYC were relatively less densely sampled than NYC boroughs, which might lead to an underestimation of the circulation of viral lineages in the surrounding areas. A denser sampling in areas surrounding NYC could help clarifying the actual extent of local dispersal of viral lineages in these areas.

In the second part of our study, we focused on the major clade identified above, as well as on the NYC area for which we had a denser sampling coverage. While the analytical pipeline applied in the first part of the study enables fast phylogeographic investigations of the dispersal history of viral lineages, it does not account for the statistical uncertainty related to the inference of tree topologies. When working on closely related genomic sequences, phylogenetic uncertainty can however be non-negligible. To accommodate this source of uncertainty together with the effects of sampling intensity differences among boroughs, we worked on downsampled sets of viral genomes and performed joint Bayesian phylogeographic inferences, i.e. joint inferences of phylogenetic trees and ancestral node locations. Specifically, we constructed ten downsampled subsets by randomly sampling sequences from NYC boroughs in proportion to their cumulative number of new COVID-19 hospitalizations until the most recent sampling date (May 10, 2020; see the Methods section for further detail). With this procedure, we thus aimed (i) to construct and analyze data subsets that are amenable for joint Bayesian phylogeographic inference, and (ii) to explicitly mitigate sampling bias by subsampling NYC boroughs according to their relative importance during the epidemiological phase under investigation. Failing to correct for sampling bias can potentially lead to artifactual outcomes in the phylogeographic reconstructions [10,11]. For each of the ten subsets, we performed both a continuous [9] and a discrete [8] phylogeographic inference. Whereas the former method allows for spatially-explicit reconstruction of the dispersal history of viral lineages, the latter enables inference of the lineage transition events among discrete locations. In the present case, we considered each NYC borough as a distinct discrete location.

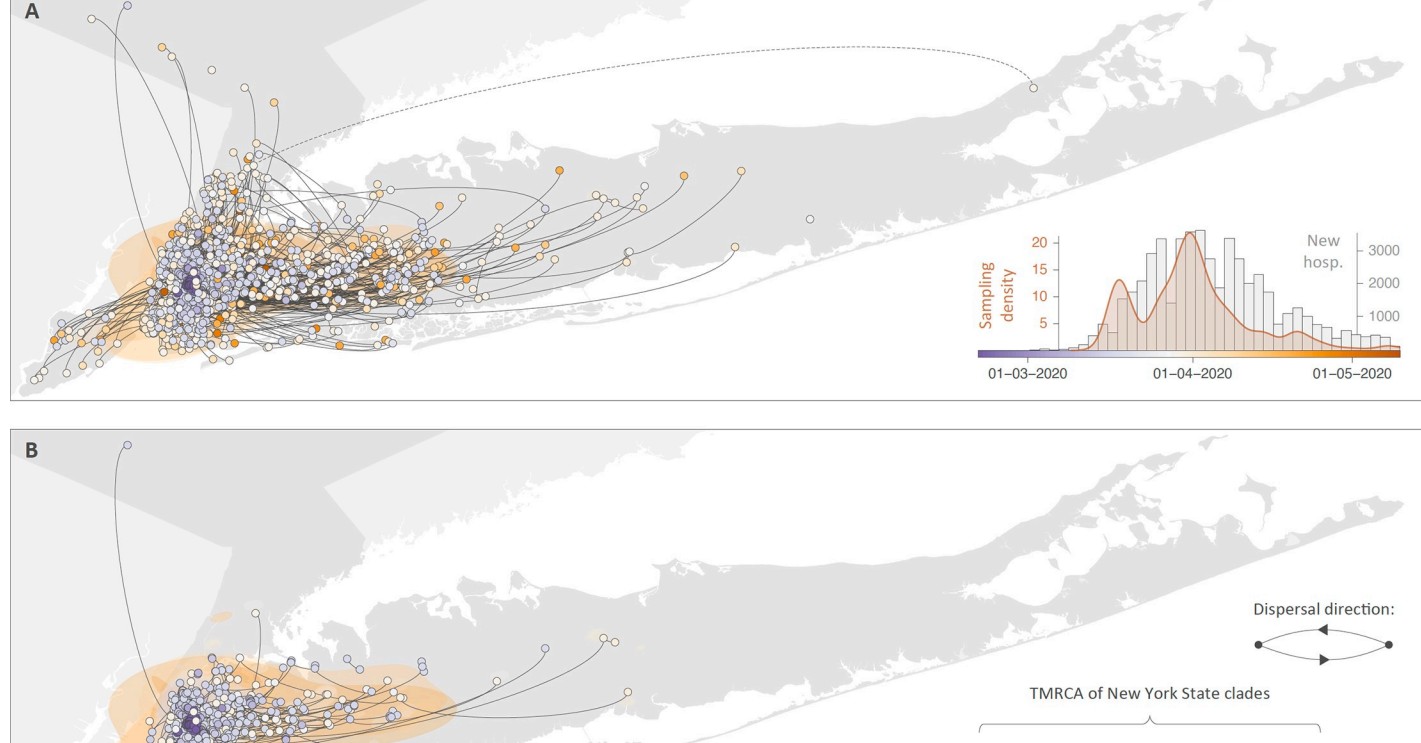

**Fig 1. Preliminary continuous phylogeographic reconstruction of the dispersal history of SARS-CoV-2 lineages in New York State. A.** Continuous phylogeographic reconstruction performed along each New York State clade identified by the initial discrete phylogeographic analysis (S1 Fig). For each clade, we mapped the maximum clade credibility (MCC) tree and overall 80% highest posterior density (HPD) regions reflecting the uncertainty related to the phylogeographic inference. MCC trees and 80% HPD regions are based on 1,000 trees subsampled from each post burn-in posterior distribution. Dispersal direction of viral lineages is indicated by the edge curvature, MCC tree nodes are colored according to their time of occurrence, and 80% HPD regions were computed for successive time layers and then superimposed using the same color scale reflecting time. Continuous phylogeographic reconstructions were only performed along New York State clades linking at least three sampled sequences for which the geographic origin was known. Besides the phylogenetic branches of MCC trees obtained by continuous phylogeographic inference, we also mapped sampled sequences belonging to clades linking less than three geo-referenced sequences. Furthermore, when a clade only comprises two geo-referenced sequences, we highlighted the phylogenetic link between these two sequences with a dashed curve connecting them (only one is visible). **B.** Fig 1B corresponds to Fig 1A but for the sake of clarity, this time we only reported internal branches and nodes of the MCC tree. In this figure, we also report the temporal distribution of the estimated time to the most recent common ancestor (TMRCA) of each circulating clade resulting from a distinct introduction event into New York State. Base layer for the maps has been obtained from https://www.census.gov.

Overall, the ten replicated continuous phylogeographic reconstructions display the same dispersal pattern, in which peripheral samples are mostly connected to the geographic center of the study area located around the northern parts of Brooklyn and Queens (Fig 2). Because continuous phylogeographic reconstructions are not necessarily easy to interpret visually, we also summarized lineage dispersal events among and within NYC boroughs (S2 Fig), which indicates that Queens, and to some extent Brooklyn, acted as main transmission hubs, here defined as areas with relatively higher local circulation that tended to act as source locations for viral lineages into the surrounding boroughs. Finally, we have also compared those continuous phylogeographic reconstructions with the same analyses performed under an alternative diffusion model allowing the assessment and inference of a directional trend [12]. The resulting continuous phylogeographic analyses did not reveal any evidence of a directional trend and lead to highly similar phylogeographic reconstructions (S3 Fig).

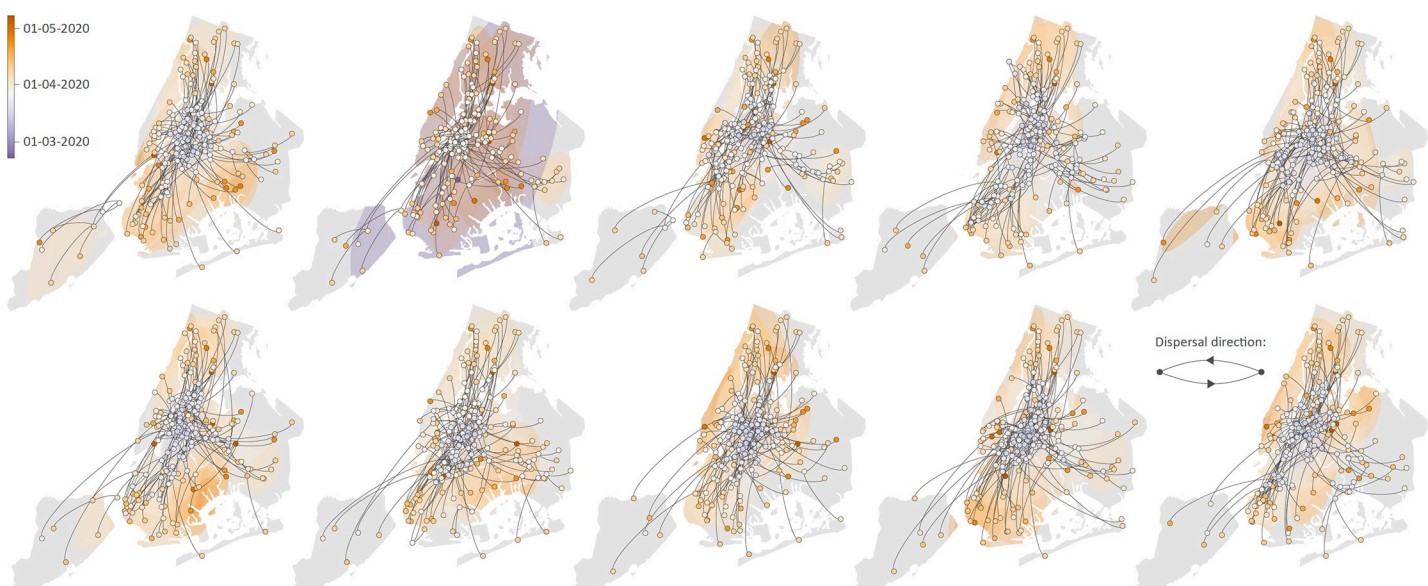

**Fig 2. Continuous phylogeographic analyses of the main SARS-CoV-2 clade circulating in New York City (NYC) during the first epidemic wave.** Each map corresponds to a distinct replicated phylogeographic inference based on a random subset of genomic sequences obtained after having subsampled NYC boroughs according to their cumulative number of new hospitalizations until the most recent sample (see the Methods section for details). For each replicated analysis, we mapped the maximum clade credibility (MCC) tree and overall 80% highest posterior density (HPD) regions reflecting the uncertainty related to the Bayesian phylogeographic inference. MCC trees and 80% HPD regions are based on 1,000 trees sampled from each post burn-in posterior distribution. Dispersal direction (anti-clockwise) of viral lineages is indicated by the edge curvature, MCC tree nodes are colored according to their time of occurrence, and 80% HPD regions were computed for successive time layers and then superimposed using the same color scale reflecting time. See also S2 Fig for an alternative visualization summarizing lineage dispersal events among and within NYC boroughs, and S3 Fig for the alternative yet consistent continuous phylogeographic reconstructions based on a diffusion model allowing the assessment and inference of a directional trend. Base layer for the maps has been obtained from https://www.census.gov.

To further explore the dispersal dynamics of viral lineages among NYC boroughs, we also employed a complementary discrete phylogeographic inference approach. While not spatially-explicit, this alternative approach provides a more direct focus on transition events among discretized areas, as well as on their statistical support. As was the case for the continuous analyses, we performed a discrete phylogeographic inference for each of the ten downsampled data sets, and then summarized lineage dispersal events among and within NYC boroughs (Fig 3). Of the ten replicated discrete phylogeographic analyses, two inferred Brooklyn and the Bronx as central transmission hubs, whereas the remaining eight were congruent with the continuous analyses in inferring Queens as the main transmission hub during the first phase of the NYC COVID-19 epidemic. The lack of full congruence among the ten replicates is likely due to the relatively limited number of samples considered in each replicated analysis. Indeed, the random selection of those samples can potentially lead to different dispersal inferences illustrating the impact of the sampled phylogenetic diversity in shaping the inferred dispersal history of sampled lineages. From a methodological point of view, these results further underline the importance of explicitly dealing with heterogeneous sampling and the consequent need for reliable and robust replicate subsampling procedures for investigating the impact of small subsets of samples on phylogeographic outcomes.

Overall, our results further confirm the epidemiological importance of the borough of Queens during the first phase of the epidemic [1]: this borough was both the scene of a relatively higher local circulation of viral lineages and the origin of a relatively high number of lineage transition events towards neighboring NYC boroughs. With more than 2.5 and 2.2 million people, Brooklyn and Queens are respectively the most populated boroughs of NYC

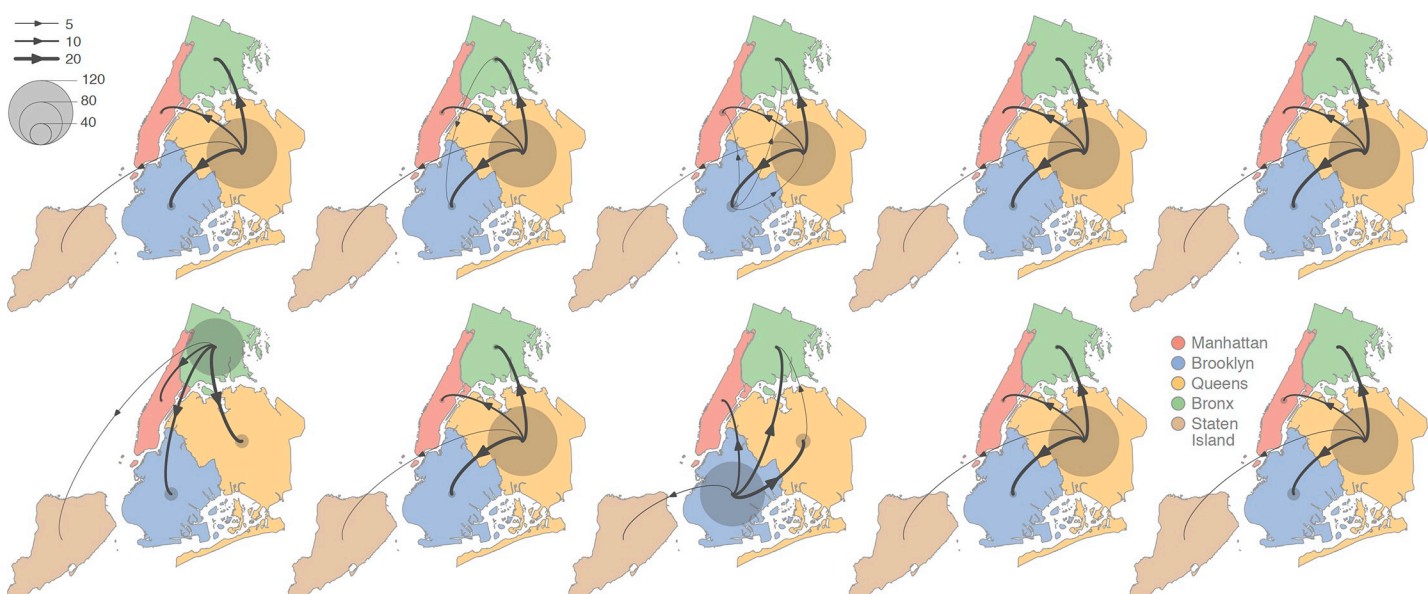

**Fig 3. Schematic overview of the discrete phylogeographic analyses of the main SARS-CoV-2 clade circulating in New York City (NYC) during the first epidemic wave.** These maps schematize the outcome of each replicated discrete phylogeographic inference based on a random subset of genomic sequences (see the Methods section for further detail). We here report the number of lineage dispersal events inferred among (arrows) and within (transparent grey circles) NYC boroughs, both measures being averaged over 1,000 posterior trees sampled from each posterior distribution. See also S4 Fig for the estimated Markov jumps among NYC boroughs, an alternative yet congruent representation of the lineage transition events inferred by discrete phylogeographic analyses. Base layer for the maps has been obtained from https://www.census.gov.

(www.census.gov), with ~35% of their workers daily commuting to Manhattan (nycfuture. org). As in many other cities around the world, important commuting activity radiating out of central city areas likely played an important role in disseminating viral lineages throughout the state. However, commuting workers are likely not the only drivers of SARS-CoV-2 dissemination across the city. Unfortunately, the scarcity of testing in the early phase of the pandemic, especially in certain neighborhoods, makes it difficult to discuss the potential drivers of the dissemination of SARS-CoV-2 within the city and suburban areas. One important point is that Black and Hispanic New Yorkers were hit disproportionately by the pandemic, and the boroughs of Queens, Brooklyn, and the Bronx have a higher proportion of Black and Hispanic residents compared to the island of Manhattan [3].

Large-scale viral genome sequencing enables inference of the dispersal history of SARS-CoV-2 lineages [13,14], which is a key step for understanding COVID-19 epidemiological dynamics. In addition, molecular epidemiological analyses can also focus on target mutations or variants of interest/concern. To illustrate this, we here identified and mapped several spike protein mutations detected during the first NYC epidemic wave (Fig 4). It is now more important than ever to track the spread of variants/mutations of concern. Indeed, the recent emergence of the 501Y.V1 (lineage B.1.1.7), 501Y.V2 (lineage B.1.351), and 501Y.V3 (lineage P.1) variants first identified in the United Kingdom, South Africa, and Brazil, respectively, have pushed many countries to enhance their genomic surveillance of the SARS-CoV-2 diversity across their territory. As modelling analyses have estimated that variants like 501Y.V1 are associated with a higher transmissibility [15,16], monitoring their colonization should be a top priority in the upcoming months. Of interest, as of early January 2021, the 501Y.V1 variant has also been identified in different US states including New York. Among our first wave sequences from New York, the N501Y spike protein mutation appeared in only one sequence (Fig 4). In contrast to the 501Y.V1-V3 variants, the N501Y mutation was not associated with any described

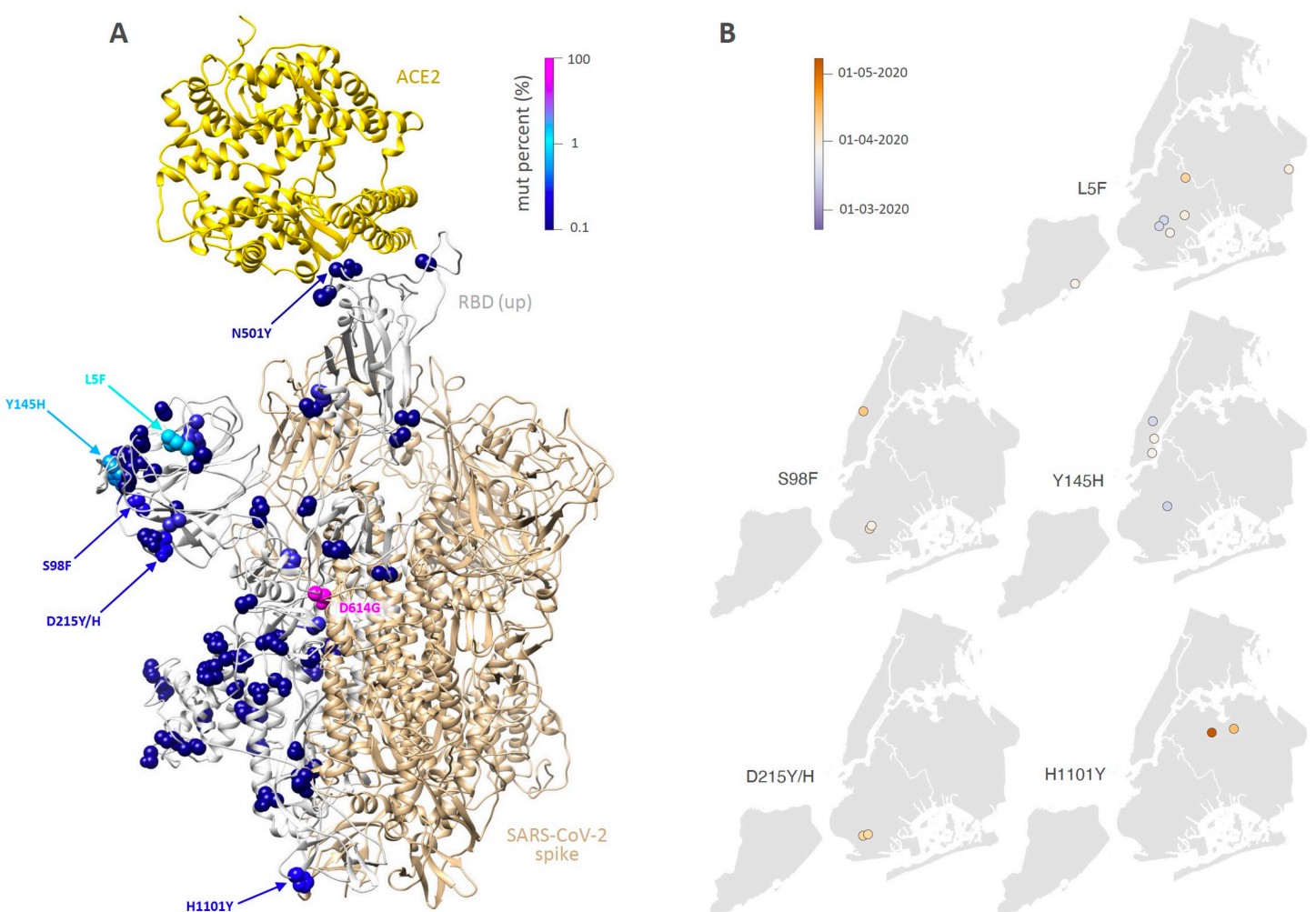

**Fig 4. Visualization and spatio-temporal distribution of spike mutations observed in our NYC study sequences. A.** 3D structure of a partially "open" SARS-CoV-2 spike trimer in ribbon representation (gold) in complex with the angiotensin-converting enzyme 2 (ACE2) receptor (https://zhanglab.ccmb.med.umich.edu/COVID-19). The monomer with the receptor-binding domain (RBD) in up position is colored gray and all spike mutations (mut) observed in our NYC data set are shown as spheres and colored according to their relative abundances (color-code in upper right corner). Spike mutations that were detected at least three times are labeled (spatio-temporal distribution investigated in panel B). Additionally, the N501Y mutation was detected only once but is known as part of the emerging 501Y.V1-V3 variants of concern. **B.** Spatio-temporal distribution of SARS-CoV-2 samples carrying spike-specific mutations. We here focus on spike mutations detected at least three times in the overall state sampling except for mutation D614G, which was the dominant phenotype, present in 95% of the samples we analyzed. Each dot corresponds to a given sample and is colored according to its collection date. Base layer for the maps has been obtained from https://www.census.gov.

variant with the reported set of mutations or amino acid deletions (the latter found in 501Y.V1), but occurred together with single nucleotide polymorphism C241T and amino acid replacements nsp2: T85I, nsp12: P323L, nsp14: S369F, S: D614G, and ORF3a: Q57H. In this context, global [17] and more local [18] phylogeographic investigations can be efficient tools to uncover their dispersal history, especially in a context where genomic surveillance will intensify in NYC and other parts of the world due to the threat of new SARS-CoV-2 variants of concern.

## Methods

### Discrete and continuous phylogeographic methods used in this study

In both parts of this study, we used two distinct yet complementary Bayesian phylogeographic methods implemented in BEAST 1.10 [19], one discrete [8] and the other continuous [9]. The

first method is a discrete character mapping approach modeled according to a continuous-time Markov chain, here characterized by a matrix of asymmetrical transition rates among sampling locations [8,20]. As such, a high transition rate to a particular location reflects a large number of transition events to that location and hence many samples from that location being present in the data set. Thus, in reconstructing dispersal dynamics, this discrete method relies on a data set with samples from all locations where infections are occurring, with the number of samples taken from each location ideally proportional to the location's incidence. The absence of samples from a particular location is interpreted as the virus not spreading to that location. While the discrete phylogeographic method requires the preliminary delimitation of discrete locations exchanging viral lineages (and hence cannot estimate ancestral locations that are not part of this predefined set), the continuous method is based on a spatially-explicit model in which internal nodes can occur in unsampled locations. This continuous method employs an anisotropic relaxed random walk (RRW) [21] and can be considered as a continuous character mapping approach treating longitude and latitude as two separate continuous traits.

## First part: preliminary phylogeographic analyses at the New York State level

The first part of this study applies a workflow [6] based on a single time-scaled phylogenetic tree, which was obtained as follows: (i) we first estimated a maximum-likelihood phylogeny using IQ-TREE 2.0.3 [22] under a general time-reversible (GTR) model of nucleotide substitution [23] with empirical base frequencies and four site rate categories [24], which was selected as the best GTR model using IQ-TREE's ModelFinder tool; (ii) we subsequently time-calibrated the resulting maximum-likelihood tree using TreeTime 0.7.4 [25], specifying a clock rate of $8 \times 10^{-4}$ substitutions per site per year (s/s/y), as in the Nextstrain workflow.

The resulting time-scaled phylogenetic tree served as a fixed evolutionary history [26] for preliminary phylogeographic analyses based on the discrete diffusion model [8] implemented in the software package BEAST 1.10 [19]. As the purpose of this first phylogeographic analysis was to delineate clades corresponding to distinct introduction events of SARS-CoV-2 lineages into New York State, we only considered two possible ancestral locations: "New York State" and "other location". We inferred posterior distributions for Bayesian phylogeographic models by employing a Metropolis-Hastings algorithm for Markov chain Monte Carlo (MCMC) simulations. Each Markov chain was simulated for $10^6$ generations and sampled every $10^3$ generations, and MCMC convergence and mixing properties were inspected using Tracer v.1.7 [27]. After discarding 10% of the sampled trees as burn-in, a maximum clade credibility (MCC) summary tree was constructed using TreeAnnotator 1.10 [19] and used to delineate clades corresponding to independent introduction events into New York State. Specifically, we considered an introduction event to be any branch in the phylogenetic tree where the location assigned to a node was "New York State" and the location assigned to its parent node in the tree was "other location".

We used the RRW diffusion model [9] available in BEAST 1.10 [19] to perform continuous phylogeographic reconstructions along clades delineated in the previous step and connecting at least three sequences sampled in New York State. Continuous phylogeographic inference requires unique sampling coordinates assigned to the tips of the tree. For each sampled genome, we retrieved geographic coordinates from a point randomly sampled within its zip code area of origin, which is the maximal level of spatial precision in the available metadata. The MCMC chain was run for $5 \times 10^8$ generations and sampled every $10^5$ generations, its convergence/mixing properties were again assessed with Tracer, and an appropriate number of

sampled trees was discarded as burn-in. The resulting sets of plausible trees were used to obtain clade-specific MCC summary trees with TreeAnnotator, and we then employed functions available in the R package "seraphim" [28] to extract spatio-temporal information embedded within posterior trees and visualize the continuous phylogeographic reconstructions.

## Second part: continuous and discrete phylogeographic analyses at the NYC level

Ten random subsets of sequences sampled in NYC and belonging to the large New York State clade were generated. Specifically, we subsampled NYC boroughs according to their cumulative number of new COVID-19 hospitalizations until the most recent sample date (Manhattan: >7,100; Brooklyn: >12,900; Queens: >15,400; the Bronx: >11,000; Staten Island: >2,100; source: https://github.com/thecityny/covid-19-nyc-data). Because the Bronx area was the proportionally least sampled borough when comparing available number of sequences to cumulative number of COVID-19 hospitalizations, the sampling intensity of this borough (2.36 sequences per 1,000 cumulated new hospitalizations) served as reference for downsampling the available number of sequences from other NY boroughs such that the number of remaining sequences of the boroughs all reflect the same sampling intensity. The downsampled data sets comprised the following number of sequences: $n$ = 17 (Manhattan), 31 (Brooklyn), 36 (Queens), 26 (the Bronx), and 5 (Staten Island). We acknowledge that the cumulative number of new hospitalizations is not the absolute metric to measure the local epidemic's intensity. However, we used this metric instead of the number of positive cases because the latter is by essence impacted by the testing effort and strategy, both of which evolved through time in New York State over the course of the early epidemic there.

For each downsampled data set, we performed both a continuous and a discrete phylogeographic analysis using the respective diffusion models implemented in BEAST 1.10 [19]. For the continuous analysis, we used a Cauchy distribution to model the among-branch heterogeneity in diffusion velocity [9], and for the discrete analysis, we used the Bayesian stochastic search variable selection (BSSVS) approach to identify the best-supported lineage transitions events between NYC boroughs [8]. Additionally, BSSVS enables to determine which transition rates are zero depending on the signal in the data, and hence to estimate the number of transitions that appropriately explains the viral diffusion process. We again employed a Metropolis-Hastings algorithm for MCMC chains that were run for sufficiently long (from $25 \times 10^7$ to $10^9$ iterations for the continuous analyses and from $12 \times 10^7$ to $6 \times 10^8$ iterations for the discrete analyses) to reach adequate ESS values as estimated by the program Tracer 1.7 [27], sampling trees every $5 \times 10^4$ iterations and discarding 10% of sampled trees as burn-in. For both kinds of phylogeographic analyses, we modeled the substitution process according to a GTR+Γ parametrization [23] and specified a flexible skygrid model as the tree prior [29]. The lack of temporal signal associated with the subsets of sequences (confirmed using the root-to-tips regression approach implemented in TempEst 1.5.3 [30]) was accommodated by (i) fixing the substitution rate to an independent estimate ($8.431 \times 10^{-4}$ s/s/y) and (ii) constraining the root age to the age of the corresponding clade in the time-scaled phylogenetic tree inferred in the first part of this study. The evolutionary rate estimate here corresponds to the posterior mean estimated with BEAST 1.10 under a GTR+Γ substitution model, a strict clock, and an exponential growth model, for a subsample of 302 SARS-CoV-2 sequences from the original alignment, longitudinally sampled through time to maximize temporal coverage.

While the RRW model is an anisotropic diffusion model that has demonstrated the ability to infer non-central locations of internal nodes [21], it has been recognized that in some situations it can inadequately model a diffusion process characterized by directional trends [12]. To

assess this potential issue, we re-performed the ten replicated continuous phylogeographic analyses using a recently developed relaxed directional random walk (RDRW) model [12]. Compared to the RRW model in which longitude and latitude are treated as continuous traits modelled according to a time-scaled mixture of Brownian diffusion processes with a zero-mean displacement along each phylogenetic branch, the RDRW model relaxes the assumption of a zero-mean displacement along phylogenetic branches, which enables inference of directional trends associated with the dispersal of lineages.

All the post hoc analyses were performed on 1,000 trees sampled from each (post-burn-in) posterior distribution, and corresponding MCC summary trees were identified as before [19]. For the continuous analyses, we again used the R package "seraphim" for mapping MCC trees and associated highest posterior density (HPD) regions reflecting the uncertainty related to the Bayesian phylogeographic inference. For the discrete analyses, we used and compared two different approaches to report the results of the phylogeographic reconstructions. First, we reported Markov jumps as estimated by the BSSVS analyses and supported by Bayes factor (BF) values >3, which correspond to at least "positive" statistical support following the scale of interpretation defined by Kass & Raftery [31]. BF support was approximated in two ways: the standard BF support [8] and the adjusted BF support that takes into account the relative abundance of samples by location [32], the latter being based on a methodology similar to the tip-date randomization test for temporal signal [33]. Second, we reported lineage transition events averaged across the sets of post burn-in trees, which also allows reporting lineage transition events occurring within discretized locations. Structural analyses were performed using the program Chimera 1.15 [34].

## Supporting information

**S1 Fig. Time-scaled phylogenetic tree in which we identified SARS-CoV-2 phylogenetic clades introduced in New York State during the first epidemic wave.** We delineated those clades by performing a discrete phylogeographic reconstruction along a time-scaled phylogenetic tree while only considering two potential ancestral locations: "New York State" and "other location". We identified a minimum number of 116 lineage introductions (95% HPD interval = [107–127]), which showcases the relative importance of external introductions considering the number of sequences sampled in our labs during the first wave in New York State (*n* = 828). On the phylogenetic tree, lineages circulating in New York State are highlighted in purple, and larger purple nodes correspond to the most ancestral node of each clade. (*) refers to the most recent common ancestor inferred for the major New York State clade on which we focused in the second part of this study dedicated to integrated continuous and discrete phylogeographic inference based on downsampled subsets of SARS-CoV-2 genomic sequences. In the lower-left corner, we report the distribution of the sizes of New York State clades. (**) For ease of visualization, the major state clade comprising 596 genomic sequences is not shown in the histogram.
(PDF)

**S2 Fig. Schematic overview of the continuous phylogeographic analyses of the main SARS-CoV-2 clade circulating in New York City (NYC) during the first epidemic wave.** On these maps schematizing the outcome of each replicated continuous phylogeographic analysis (Fig 2), we report the number of lineage dispersal events inferred among (arrows) and within (transparent grey circles) NYC boroughs, with both measures being averaged over 1,000 posterior trees sampled from each posterior distribution. Base layer for the maps has been obtained from https://www.census.gov.
(PDF)

**S3 Fig. Alternative continuous phylogeographic analyses of the main SARS-CoV-2 clade circulating in New York City (NYC) during the first epidemic wave.** Contrary to the analyses reported in Fig 2 and obtained using the relaxed random walk (RRW) model, these alternative analyses were obtained using the relaxed directional random walk (RDRW) diffusion model, and led to phylogeographic reconstructions highly similar to the ones obtained with the RRW model (Fig 2). With this alternative RDRW diffusion model, maximum clade credibility (MCC) trees have, on average, fewer than 2% of their branches associated with significant latitudinal trends and fewer than 2% of their branches associated with significant longitudinal trends. Furthermore, RDRW analyses did not infer any global directional trend. As in Fig 2, each map corresponds to a distinct replicated phylogeographic inference based on a random subset of genomic sequences obtained after having subsampled NYC boroughs according to their cumulative number of new hospitalizations until the most recent sample (see the Methods section for details). For each replicated analysis, we mapped the maximum clade credibility (MCC) tree and overall 80% highest posterior density (HPD) regions reflecting the uncertainty related to the Bayesian phylogeographic inference. MCC trees and 80% HPD regions are based on 1,000 trees sampled from each post burn-in posterior distribution. Dispersal direction (anti-clockwise) of viral lineages is indicated by the edge curvature, MCC tree nodes are colored according to their time of occurrence, and 80% HPD regions were computed for successive time layers and then superimposed using the same color scale reflecting time. Base layer for the maps has been obtained from https://www.census.gov.
(PDF)

**S4 Fig. Well-supported Markov jumps inferred by discrete phylogeographic inferences dedicated to the main SARS-CoV-2 clade circulating in New York City (NYC) during the first epidemic wave.** Each map corresponds to a distinct replicate discrete phylogeographic inference based on a random subset of genomic sequences obtained after having subsampled NYC boroughs according to their overall number of positive cases recorded until the most recent sample. As an alternative visualization to the average number of lineage transition events reported in Fig 3, we here report supported Markov jumps among NYC boroughs. Markov jumps are either supported by standard (**A**) or adjusted (**B**) Bayes factor values higher than 3, which correspond to positive support according to the scale of interpretation defined by Kass & Raftery (1995). Base layer for the maps has been obtained from https://www.census.gov.
(PDF)

## Author Contributions

**Conceptualization:** Simon Dellicour, Guy Baele, Ralf Duerr, Adriana Heguy.

**Data curation:** Samuel L. Hong, Sitharam Ramaswami, Ralf Duerr, Adriana Heguy.

**Formal analysis:** Simon Dellicour, Samuel L. Hong, Bram Vrancken, Mandev S. Gill, Ralf Duerr.

**Funding acquisition:** Ralf Duerr, Adriana Heguy.

**Investigation:** Simon Dellicour, Samuel L. Hong, Bram Vrancken, Antoine Chaillon, Mandev S. Gill, Matthew T. Maurano, Sitharam Ramaswami, Paul Zappile, Christian Marier, Gordon W. Harkins, Guy Baele, Ralf Duerr, Adriana Heguy.

**Methodology:** Simon Dellicour, Samuel L. Hong, Bram Vrancken, Mandev S. Gill, Guy Baele.

**Project administration:** Ralf Duerr, Adriana Heguy.

**Resources:** Simon Dellicour, Samuel L. Hong, Bram Vrancken, Antoine Chaillon, Matthew T. Maurano, Sitharam Ramaswami, Paul Zappile, Christian Marier, Gordon W. Harkins, Guy Baele.

**Software:** Simon Dellicour.

**Supervision:** Adriana Heguy.

**Validation:** Simon Dellicour, Samuel L. Hong, Bram Vrancken, Antoine Chaillon, Mandev S. Gill, Gordon W. Harkins, Guy Baele, Ralf Duerr.

**Visualization:** Simon Dellicour, Ralf Duerr.

**Writing – original draft:** Simon Dellicour.

**Writing – review & editing:** Simon Dellicour, Samuel L. Hong, Bram Vrancken, Antoine Chaillon, Mandev S. Gill, Matthew T. Maurano, Sitharam Ramaswami, Paul Zappile, Christian Marier, Gordon W. Harkins, Guy Baele, Ralf Duerr, Adriana Heguy.

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
