## [Decision Letter · Decision Letter 0]

28 Feb 2021

Dear Dr. Dellicour,

Thank you very much for submitting your manuscript "Dispersal dynamics of SARS-CoV-2 lineages during the first epidemic wave in New York City" for consideration at PLOS Pathogens. As with all papers reviewed by the journal, your manuscript was reviewed by members of the editorial board and by several independent reviewers. In light of the reviews (below this email), we would like to invite the resubmission of a significantly-revised version that takes into account the reviewers' comments.

We apologize for the delay in review. Both reviewers thoroughly evaluated the manuscript. While reviewer 1 is generally positive, there is some concern that the findings might be relatively limited given the focus on just one introduction and the lack of conclusions that can be drawn about the others. Reviewer 2 raises a number of concerns about how the analysis is presented and perhaps issues that can affect overall interpretation. Given this, we would be willing to consider a suitably revised manuscript. But it would be necessary to address these comments in full. A revised manuscript would likely go out for review again, likely to the same reviewers.

We cannot make any decision about publication until we have seen the revised manuscript and your response to the reviewers' comments. Your revised manuscript is also likely to be sent to reviewers for further evaluation.

Sincerely,

Adam S. Lauring

Section Editor

PLOS Pathogens

Adam Lauring

Section Editor

PLOS Pathogens

Kasturi Haldar

Editor-in-Chief

PLOS Pathogens

orcid.org/0000-0001-5065-158X

Michael Malim

Editor-in-Chief

PLOS Pathogens

orcid.org/0000-0002-7699-2064

We apologize for the delay in review. Both reviewers thoroughly evaluated the manuscript. While reviewer 1 is generally positive, there is some concern that the findings might be relatively limited given the focus on just one introduction and the lack of conclusions that can be drawn about the others. Reviewer 2 raises a number of concerns about how the analysis is presented and perhaps issues that can affect overall interpretation. Given this, we would be willing to consider a suitably revised manuscript. But it would be necessary to address these comments in full. A revised manuscript would likely go out for review again, likely to the same reviewers.

Reviewer's Responses to Questions

**Part I - Summary**

Reviewer #1: This is a nice paper, about using phylogeographic methods to uncover the transmission pattern of SARS-CoV-2 in New York. The authors use a set of around 800 sequences and subsample multiple times to try and mitigate spatial sampling bias. The methods used - the discrete and continous phylogeographic methods are well chosen and executed for the data and the limitations of the methods are clearly discussed.

However, this is paper focuses on spread of one main introduction to NY, and according to the data shown this is the main clade in wave 1. Presumably the other introductions didn't seed (sub)lineages long lived enough to sample and analyse, and it would be useful to include some more on these so that the difference between the successful and unsuccessful (sub)lineages can be more easily shown.

Reviewer #2: [Please see the attached PDF, which may have some typographical elements the following plain-text version misses.]

In this manuscript, the authors apply existing phylogeographic methods to SARS-CoV-2 genome sequences to obtain insight into patterns of virus transmission within the greater New York conurbation during the early phases of the COVID-19 pandemic. They first perform a single-tree analysis and then expand this to an analysis that incorporates phylogenetic uncertainty. In both of these, they employ both a continuous-space and then a discrete-space (metapopulation) model.

While the subject is interesting and important, it is impossible to adequately judge the significance of the conclusions because the methods are only perfunctorily characterized and described. In particular, the authors seek to justify their methods via a mere appeal to authority (saying, e.g., that the workflows are "standard" per other authors). There is nothing in phylogeographic analysis, however, that deserves to be considered standard. Indeed, even some common practices in this sort of analysis are questionable. Be this as it may, inasmuch as it is not even possible to *state* the conclusions of the paper without reference to the methods, it is essential that sufficent methodological detail be provided that the reader can understand the conclusions for what they are. Actually, more is required, since without an appreciation of the assumptions that have gone into the methods, one cannot judge which aspects of the conclusions are most strongly supported by the data, which are consequences of arbitrary but reasonable model assumptions, and which (if any) are consequences of invalid assumptions. Nor is it only the credibility of the conclusions that is influenced by the methods; methodological details shape one's appreciation of the conclusions' import as well. Finally, because the methodological details are so intimately connected with the data and the conclusions, it is essential that they be included in the paper in much more detail than they are currently given.

Here is an example. With respect to the continuous-space model, inasmuch as the model (including the prior) is isotropic, the effect will be to pull the physical locations attributed to phylogenetic nodes toward the center of the space. Accordingly, I suspect that the identification of the Greenpoint/Long Island City area as a locus of high phylogenetic centrality may be a more a consequence of its geographic centrality on the regional map than anything else. At any rate, this suspicion deserves exploration: how robust is that specific conclusion to varying priors?

I trust that, in responding to this and related criticism, the authors will feel no temptation to take refuge in nominally flat or so-called "uninformative" priors. As is well known, this is a canard: there is no such thing as an absolutely flat prior and even uniform priors contribute information to the posteriors. Nor should this be an instance where it will be sufficient to perform a bit of perfunctory fiddling with priors and other model assumptions, followed by verification that the conclusions remain superficially unchanged. Rather, the authors should ask themselves the question: Within the bounds of what would be considered permissible in a Bayesian analysis, how much prejudice against each of our conclusions must we build into the model to get a meaningfully different result? And also: As we build into the model increasing degrees of prejudice against our conclusions, in what ways do the latter change, either quantitatively and qualitatively? If the information in the data turns out to be so overwhelming that only extreme prejudice can suppress it, it is important to know this. On the other hand, if (as I suspect) it turns out that some aspects of the conclusions are highly sensitive to model assumptions, it is equally important to know it.

To get at these issues, a variety of incisive experiments with the model can be readily imagined. For example, does including Nassau or Westchester Counties, or northern New Jersey, with their large populations and strong economic ties to the city, affect the location of regions of high phylogenetic centrality? What happens if one drops Staten Island?

From this point of view, the discrete-space model is less obviously, but potentially equally, problematic. It is just as important to understand its assumptions in detail. For example, is it a perfectly generic, flexible migration model (with $n\\,(n-1)$ parameters), or do covariates (perhaps population sizes, commuting data, etc.) constrain the model? The remarks above concerning the importance of priors and other model assumptions apply with equal force here. Indeed, such considerations are in a sense even more critical here than they were for the continuous-space model, since it may be more difficult to identify the consequences of arbitrary model assumptions in this more complex model.

Certain others of the model assumptions appear quite arbitrary. For example, how does one justify allowing localities within New York State to play a distinguished role not shared by regions of New Jersey and Connecticut that are potentially much more strongly connected to the city in all the ways that matter for transmission? Similarly, what is the magnitude of population movements into the city from more far-flung regions via the city's airports and intercity trains during this period, relative to those from the suburbs? I do not mean to suggest that it is strictly necessary that the authors expand the geographic scope of their study. Rather, my point is that one needs to investigate the extent to which geographical distinctions made in the analysis---distinctions which are respected neither by the host nor the virus---may be responsible for certain of the quantitative or qualitative conclusions.

Finally, I anticipate that, in the course of improving their analysis, the authors may develop hypotheses as to the causal mechanisms underlying certain of the patterns they observe. The paper would be much strengthened were they to follow up on these, using additional data as appropriate. After all, by their very nature, the highly flexible, generic models of the sort employed in this paper, can say little, if anything, about the true drivers of the patterns observed. They are best viewed, therefore, as sophisticated tools for exploratory data analysis and hypothesis generation.

**Part II – Major Issues: Key Experiments Required for Acceptance**

Reviewer #1: (No Response)

Reviewer #2: See Summary above. Also:

1. ll 80ff: The authors recognize that inadequate sampling (or perhaps overly narrow data-inclusion criteria) limit their ability to resolve clusters with substantial extent beyond the boundaries of the city proper. In view of this lack of resolution, it would be more straightforward to state it as I just have rather than as a positive result ("Continuous phylogeographic reconstructions highlight that most peripheral samples are directly connected to the NYC area rather than clustered together (Fig. S2). In other words, samples collected outside NYC likely correspond to distinct dispersal events originating from the city area.") qualified by a caveat ("However, we acknowledge that areas surrounding NYC were relatively less densely sampled than NYC boroughs, which might lead to an underestimation of the circulation of viral lineages in the surrounding areas."). It seems that one could, with equal evidence and at least as great plausibility, put it the other way around.

2. l 97: "...according to their relative importance during the epidemiological phase under investigation". The authors assume that relative importance is adequately measured by proportion of hospitalizations. Are there alternative metrics that might be equally pertinent? What impacts would bias in hospitalization rates among boroughs---which have markedly different income and employment profiles, and therefore presumably different rates of insurance coverage---have on the analysis?

3. l 128: In what sense is the discrete model not spatially explicit? I interpreted this model as being what an ecologist would call a spatially-explicit metapopulation model. If I did so incorrectly, this is more testimony to the need for better description of the models.

4. Why was downsampling necessary?

5. The authors highlight the importance of heterogeneities in sampling as potentially consequential to their conclusions. They ignore, however, the many other model assumptions (most of them left unmentioned for the reader to guess at according to the degree of their experience and insight) about the magnitudes and directions of host movement fluxes, transmission rates, recovery rates, etc.

6. ll 203ff: The description of the MCMC protocols begs the question as to the extent these were checked for adequacy. It is generically the case that the algorithmic parameters used by previous studies will be inadequate, or at least sub-optimal, for a given, new MCMC analysis, even when the former appear similar in some ways to the latter. How did the authors determine that their algorithmic choices (number of MCMC iterations, number of chains, length of burn-in period, etc.) achieved convergence to the posterior and allowed adequate sampling therefrom? How did they verify convergence? How did they establish mixing? What specific MCMC variant was employed?

7. l 248: The authors assumed a strict molecular clock. To what extent did this strong prior affect their conclusions?

**Part III – Minor Issues: Editorial and Data Presentation Modifications**

Reviewer #1: There are a few minor issues that I noticed:

Data set - GISAID (NY NYUMC2-929)

Is this now public accessible data and would it make sense to list the EPI_ISL in supplementary (I notice that you have the sequences info in GitHub though).

Also, although you mention this in the methods text, I wonder if it would be useful to show something like a bar graph / table with the numbers of sequences from each discrete place per epi-week ?

Line 84 - I wonder if you are mostly looking at the one big clade; would you miss lineages spreading differently in the areas surrounding NYC especially if they are seeded from outside NY area (or outside US) ?

Line 96-101 approximate - Can you indicate the range of numbers of sequences in the subsets here.

Figures 1-3: Doing the 10 replicates is a good idea, but is there an easier way to summarise all the 10 networks on maps ? for example for Figures 2 & 3 - put all 10 on the same map with line thickness and transparency being the sum of the individual replicate lines ? And then put all 10 replicates individually in the supplementary ? and then you could put the summary maps side by side in the main text. Also for figure 1 - you could just add all the networks to one map and add transparency to the lines (so that multiple links in almost the same places will re-inforce) ?

Reviewer #2: Some of the above issues are (relatively) minor, but since addressing several of the issues I have highlighted will require substantial revision, I haven't bothered to get into truly minor details.

PLOS authors have the option to publish the peer review history of their article (what does this mean?). If published, this will include your full peer review and any attached files.

Reviewer #1: No

Reviewer #2: No
---

## [Editor Report · Decision Letter 1]

13 Apr 2021

Dear Dr. Dellicour,

Thank you very much for submitting your manuscript "Dispersal dynamics of SARS-CoV-2 lineages during the first epidemic wave in New York City" for consideration at PLOS Pathogens. As with all papers reviewed by the journal, your manuscript was reviewed by members of the editorial board and by several independent reviewers. The reviewers appreciated the attention to an important topic. Based on the reviews, we are likely to accept this manuscript for publication, providing that you modify the manuscript according to the review recommendations.

Thank you for your revision. In the interest of expediting a decision, this was handled at the editorial level. Your efforts to address these comments are much appreciated. There is one more issue that should be addressed. Reviewer 2 asked for descriptions of the assumptions, priors, and MCMC properties (e.g. length, sampling, burn in) for the analyses. It appears that they were added for the fixed tree analysis, but not for the secondary analyses. Unless I have missed it in the revised manuscript, please add this information in a second revision. Once that is complete, I should be able to render a formal editorial acceptance.

Sincerely,

Adam S. Lauring

Section Editor

PLOS Pathogens

Adam Lauring

Section Editor

PLOS Pathogens

Kasturi Haldar

Editor-in-Chief

PLOS Pathogens

orcid.org/0000-0001-5065-158X

Michael Malim

Editor-in-Chief

PLOS Pathogens

orcid.org/0000-0002-7699-2064

Thank you for your revision. In the interest of expediting a decision, this was handled at the editorial level. Your efforts to address these comments are much appreciated. There is one more issue that should be addressed. Reviewer 2 asked for descriptions of the assumptions, priors, and MCMC properties (e.g. length, sampling, burn in) for the analyses. It appears that they were added for the fixed tree analysis, but not for the secondary analyses. Unless I have missed it in the revised manuscript, please add this information in a second revision. Once that is complete, I should be able to render a formal editorial acceptance.

Reviewer Comments (if any, and for reference):

Figure Files:

Data Requirements:

Reproducibility:

References:

---

## [Editor Report · Decision Letter 2]

19 Apr 2021

Dear Dr. Dellicour,

We are pleased to inform you that your manuscript 'Dispersal dynamics of SARS-CoV-2 lineages during the first epidemic wave in New York City' has been provisionally accepted for publication in PLOS Pathogens.

Best regards,

Adam S. Lauring

Section Editor

PLOS Pathogens

Adam Lauring

Section Editor

PLOS Pathogens

Kasturi Haldar

Editor-in-Chief

PLOS Pathogens

orcid.org/0000-0001-5065-158X

Michael Malim

Editor-in-Chief

PLOS Pathogens

orcid.org/0000-0002-7699-2064
---

## [Editor Report · Acceptance letter]

29 Apr 2021

Dear Dr. Dellicour,

We are delighted to inform you that your manuscript, "Dispersal dynamics of SARS-CoV-2 lineages during the first epidemic wave in New York City," has been formally accepted for publication in PLOS Pathogens.

Best regards,

Kasturi Haldar

Editor-in-Chief

PLOS Pathogens

orcid.org/0000-0001-5065-158X

Michael Malim

Editor-in-Chief

PLOS Pathogens

orcid.org/0000-0002-7699-2064